# Wnt-11 Expression Promotes Invasiveness and Correlates with Survival in Human Pancreatic Ductal Adeno Carcinoma

**DOI:** 10.3390/genes10110921

**Published:** 2019-11-11

**Authors:** Dafydd A. Dart, Damla E Arisan, Sioned Owen, Chunyi Hao, Wen G. Jiang, Pinar Uysal-Onganer

**Affiliations:** 1Cardiff China Medical Research Collaborative, Institute of Cancer and Genetics, Cardiff University Henry Wellcome Building, Heath Park, Cardiff CF14 4XN, UK; a.dart@imperial.ac.uk (D.A.D.); s.owen@southwales.ac.uk (S.O.); JiangW@cardiff.ac.uk (W.G.J.); 2Imperial College London, Hammersmith Hospital Campus, London W12 0NN, UK; 3Science and Literature Faculty, Department of Molecular Biology and Genetics, Istanbul Kultur University, Atakoy Campus, Istanbul 34156, Turkey; damlaarisan@gmail.com; 4Alfred Russel Wallace Building, Upper Glyntaff, University of South Wales, Pontypridd CF37 4BD, UK; 5Beijing Cancer Institute and Key Laboratory of Carcinogenesis & Translational Research (Ministry of Education), Peking University School of Oncology, Beijing 100142, China; haochunyi@bjmu.edu.cn; 6School of Life Sciences, College of Liberal Arts and Sciences, University of Westminster,115 New Cavendish Street, London W1W 6UW, UK

**Keywords:** Wnt-11, pancreatic ductal adenocarcinoma, epithelial-mesenchymal transition, invasion

## Abstract

Pancreatic ductal adenocarcinoma (PDAC) is one of the deadliest forms of cancer, proving difficult to manage clinically. Wnt-11, a developmentally regulated gene producing a secreted protein, has been associated with various carcinomas but has not previously been studied in PDAC. The present study aimed to elucidate these aspects first in vitro and then in a clinical setting in vivo. Molecular analyses of Wnt-11 expression as well as other biomarkers involved qRT-PCR, RNA-seq and siRNA. Proliferation was measured by MTT; invasiveness was quantified by Boyden chamber (Matrigel) assay. Wnt-11 mRNA was present in three different human PDAC cell lines. Wnt-11 loss affected epithelial-mesenchymal transition and expression of neuronal and stemness biomarkers associated with metastasis. Indeed, silencing Wnt-11 in Panc-1 cells significantly inhibited their Matrigel invasiveness without affecting their proliferative activity. Consistently with the in vitro data, human biopsies of PDAC showed significantly higher Wnt-11 mRNA levels compared with matched adjacent tissues. Expression was significantly upregulated during PDAC progression (TNM stage I to II) and maintained (TNM stages III and IV). Wnt-11 is expressed in PDAC in vitro and in vivo and plays a significant role in the pathophysiology of the disease; this evidence leads to the conclusion that Wnt-11 could serve as a novel, functional biomarker PDAC.

## 1. Introduction

Pancreatic ductal adenocarcinoma (PDAC) is the main form of pancreatic cancer and has one of the highest death rates among human cancers with a 5-year survival of less than 4% [1,2,3]. The high mortality rate is due mainly to difficulties in both diagnosis and treatment [2], as the disease is relatively symptomless until the late stages. Carbohydrate antigen (CA) 19-9 is currently the only available biomarker for PDAC; however, it is not reliable due to its poor sensitivity and specificity [4]. Moreover, Lewis antigen-negative individuals, who compose approximately 5%–10% of PDAC cases, are genetically unable to produce CA19-9 [4,5]. Recent studies have shown that the accuracy of CA19-9 may be increased by combining with other biomarkers, such as carcinoembryonic antigen (CEA) and/or transforming growth factor beta (TGFβ) superfamily member, macrophage inhibitory cytokine 1 (MIC-1) [6,7,8,9,10]. There is a clear and significant unmet need to identify viable, functional biomarkers of PDAC.

As regards therapy, ‘conventional’ treatments, such as surgery, radiation therapy, chemotherapy, and more recently, immunotherapy, so far have shown an impact on progression of PDAC, but this is still limited [11]. The most common chemotherapy regimen is FOLFIRINOX [folinic acid / 5FU /irinotecan / oxaloplatin]. Furthermore, unfortunately, not all patients would benefit from this therapy as personalised predictive biomarkers are lacking. Potential biomarkers include regulators of drug metabolism and activity, such as the enzyme of 5-FU catabolism—dihydropyrimidine dehydrogenase (DPD), and the target enzyme thymidylate synthase (TS) [12]. More recently, a number of biological/oncogenic targets, in particular, the epidermal growth factor receptor (EGFR), have received attention [13,14,15]. However, EGFR blockade and anti-angiogenic therapy trials were not successful for treatment of PDAC [11].

Without doubt, there is an urgent need to develop reliable novel biomarkers for PDAC both to aid diagnosis and to serve as therapeutic targets. One such promising functional biomarker is Wnt-11, a member of a family of secreted proteins that regulate tissue growth and morphogenesis during development. It has previously been reported that Wnt-11 also plays a significant pathophysiological role in major carcinomas, e.g., cancers of prostate, cervical, ovarian and colon and controls neuroendocrine differentiation [16,17,18,19]. Wnt-11 is downstream of TGFβ—shown previously to be one of the triggers of epithelial-mesenchymal transition (EMT) and chemoresistance [20,21,22]. TGFβ signalling has been shown to be involved in PDAC progression, promoting genomic instability, neoangiogenesis, immune evasion, cell motility and metastasis in late-stage disease [23,24]. Moreover, TGFβ ligands are commonly overexpressed in PDAC and can promote EMT, enabling cancer cells to gain invasive properties and initiating metastasis [25,26]. Another frequently observed characteristic of aggressive carcinomas is expression of molecular markers normally associated with neurones (e.g., [27,28,29]). However, whether Wnt-11 controls EMT and/or expression of neuronal markers (NEMs) in PDAC is unknown. It is also not known if Wnt-11 is involved in cellular invasiveness in PDAC and how it may relate to survival of patients. The present study aimed to elucidate these aspects first in vitro and then, in a clinical setting, in vivo.

## 2. Materials and Methods

### 2.1. Cell Lines

Three PDAC cell lines, BxPC-3, MiaPaca-2 and Panc-1, were selected to study and purchased from ATCC. The cells maintained at 37 °C and 5% CO_2_ and were grown either in Dulbecco’s Modified Eagle Medium (DMEM) with high glucose and stable glutamine (for MiaPaCa-2 and Panc-1) or Roswell Park Memorial Institute medium (RPMI) with high glucose and stable glutamine (for Bx-PC3) supplemented with 10% FBS. 

### 2.2. Patient Samples

Fresh frozen PDAC tissues (*n* = 111), with matched normal adjacent tissues from the same patients, were collected following surgical resection at the Beijing Cancer Hospital and were stored. Clinicopathological factors, such as age, sex, histological type, TNM stage, and lymph node metastasis were recorded and stored in the patients’ database. The local ethics committee approved all the protocols and consent was obtained from the patients. Wnt-11 mRNA levels were quantified using real-time qPCR (qRT-PCR), with cytokeratin 19 (CK19) as the normalizing (“house-keeping”) gene for the tissue samples. Fresh-frozen pancreatic adenocarcinoma tissues (*n* = 202), along with matched normal tissue from the same patients, were collected immediately after surgical resection at the Beijing Cancer Hospital and were stored at the Tissue Bank of Peking University Oncology School. Clinicopathological factors such as age, sex, histological type, TNM stage, and lymph node metastasis, were recorded and stored in the patient database. All protocols were reviewed and approved by the local Ethics Committee (MTA01062008) and consent was obtained from the patients. Patients were recruited between January 2002 and December 2009 and were routinely followed up.

### 2.3. Isolation of RNA, cDNA and qRT-PCR

mRNA was isolated using an mRNA extraction kit according to the manufacturer’s instructions (Qiagen, Germantown, MD, USA). RNA quality and quantity were assessed NanoDrop Spectrophotometer at 260 nm and 280 nm absorbance. cDNA was generated and the following genes were studied (corresponding primer sequences given in references in parentheses): Wnt-11, NSE, Hes6, Neuro D [16]; Nanog [30]; Vim, Snail, Twist, E-cadherin (CDH1) [31]. qRT-PCR analysis was done using Taq SYBR Green premix (Qiagen, Germantown, MD, USA) and the following conditions were used: 95 °C for 15 minutes, 40 cycles at 95 °C for 15 seconds, 60 °C for 1 minute and a dissociation stage (95 °C for 15 seconds, 60 °C for 1 minute, 95 °C for 15 seconds, 60 °C for 15 seconds). Relative levels of mRNA expression were calculated using the Comparative CT/2^−ΔΔCT^ method [32] with RNA polymerase II (RPII) as the house-keeping gene for the in-cell-line-based studies [33]. 

### 2.4. RNA Sequencing

Total RNA was isolated from cells by using Trizol (Sigma, Gillingham, Dorest. UK). The reverse transcription to cDNA was performed with IonAmpliseqTM Transcriptome Human Gene Expression kit (Life Technologies). We then used the Human Gene Expression Core Panel primer set (Life Technologies) to prepare small amplicon gene expression libraries targeting 20,000 genes (95% of the RefSeq gene database). The cDNA amplicon libraries (125–300 bp) were ligated to adapters and amplified using IonXpress RNA-seq barcoded primers (5′). cDNA libraries were clonally amplified using an Ion PI template OT2 200 kit (Life technologies, USA) on an Ion OneTouch2 system (Life technologies) as per the manufacturer’s instructions. Samples were processed using the Ion Proton 200 sequencing kit and loaded onto a P1 chip and sequenced on an Ion Proton (Life technologies) using default parameters (single-end, forward sequencing). Base calling, adaptor trimming, barcode deconvolution, alignment and Ampliseq gene expression analysis was performed on Torrent Suite version 3.6 (Life technologies).

### 2.5. siRNA

Panc-1 cells were transfected with siRNA and non-targeting control (Dharmacon, Lafayette, CO, USA). siRNA transfections were performed according to the manufacturer’s protocols using DharmaFECT 2 (Dharmacon, Lafayette, CO, USA). Briefly, Panc-1 cells were seeded in 6-well plates for 48 hours before transfection using ON-TARGETplus SMARTpool WNT-11 siRNA or ON-TARGETplus non-targeting pool (Dharmacon, Lafayette, CO, USA). Transfected cells were used for RNA extraction (after 48 h) or cell invasion and immunostaining assays (72 h).

### 2.6. Invasion and Proliferation Assays

Details of cell invasion assay were as reported previously [16]. Briefly, 5 × 105 cells (transfected or control) were plated on Matrigel-coated transwell filters (BD Biosciences) in a chemotactic gradient of 1:10% FBS. After 16 hours, the total number of invaded cells was determined by MTT assay and this was confirmed by crystal violet assay. In parallel, the same number of cells plated and incubated for 16 h to determine the effect of the cell proliferation by MTT assay. Invasion and proliferation were presented as “Invasion” and “Proliferation”, i.e., percentage (%) of the readings for invaded cells/original cell number.

### 2.7. Immunostaining

Immunocytochemistry was performed as described previously [26]. Wnt-11 silenced Panc-1 and control (cell transfected with scrambled siRNA) cells were fixed in 4% paraformaldehyde for 15 minutes, blocked with 5% BSA/PBS for 30 minutes and incubated with goat anti-Wnt-11 (1:20) overnight at 4 °C. Alexa Fluor® 488 goat anti-mouse IgG (H + L) (Invitrogen) was used as secondary antibody (1:500, 60 minutes at room temperature) and nuclei were counterstained using 1.0 µM TO-PRO-3 (Invitrogen). 

### 2.8. Survival Analysis

Survival data was obtained on 22nd of Oct 2019 from Human Protein Atlas website: https://www.proteinatlas.org/ENSG00000085741-WNT11/pathology/pancreatic+cancer#Location Wnt-11 expression of total 176 patients was analysed, (low expression; *n* = 121 and high expression; *n* = 55) and Kaplan–Meier survival estimators examined the prognosis of each group of patients, and the survival outcomes of the two groups were compared by log-rank tests [34]. 

### 2.9. Data Analysis

All data were analysed as means ± standard errors. Statistical significance was determined using Student’s *t*-test or ANOVA with Newman–Keuls post-hoc analysis, as appropriate. Results were considered significant for *p* < 0.05. 

## 3. Results

Three sets of results are presented. First, we describe the Wnt-11 mRNA profile of three different PDAC cell lines and using siRNA, we demonstrate that Wnt-11 controls the expression of several biomarkers associated with metastasis. Secondly, we show that silencing Wnt-11 expression results in significant inhibition of cellular invasiveness. Thirdly, we demonstrate that Wnt-11 mRNA levels were significantly higher in clinical samples of PDAC compared with matched normal tissues. Overall, these data suggest that Wnt-11 is involves in the pathophysiology of PDAC.

### 3.1. Wnt-11 mRNA Expression Profiling

Wnt-11 mRNA expression in three different PDAC cell lines (BxPC-3, Panc-1 and MiaPaCa-2) was quantified using qRT-PCR. The BxPC-3 cell, which has the most epithelial properties, expressed relatively the least Wnt-11 mRNA and this was used as a “baseline’ (Figure 1A). Expression was significantly higher in MiaPaca-2 cells (460 + 11% cf. BxPC-3 cells; *n* = 6; *p* < 0.01; Figure 1A). Panc-1 cells expressed relatively the highest level (2600 + 15 % cf. BxPC-3 cells; *n* = 6; *p* < 0.01; Figure 1A; statistical significance was determined using Student’s *t*-test). RNA sequencing confirmed that Wnt-11 mRNA expression in Panc-1 cells was higher than MiaPaca-2 cells (Appendix A). Similar increases were found for NSE, Hes6, Nanog, Snail and Twist (Appendix A). In the following in vitro experiments, the Panc-1 cell line was adopted as a model due to its high Wnt-11 expression.

### 3.2. Effects of Silencing Wnt-11 on Biomarkers Associated with Metastasis

Silencing Wnt-11 in Panc-1 cells led to a reduction in mRNA expression to 22 + 2% of the control transfection, (*n* = 6; *p* < 0.05; Figure 1A). This was confirmed qualitatively at the protein level by immunocytochemistry (Figure 1B,C). We then determined whether Wnt-11 was involved in expression of biomarkers associated with metastasis (Figure 2). Two sets of such biomarkers were studied, those relating to neuronal characteristics and EMT [35,36]. As regards NEMs, Wnt-11 silencing led to decreases of the following (percentage reduction according to the control transfection that was used as a ‘baseline’ and presented as 100% and statistical significance was determined by using Student’s *t*-test): NSE (53 + 2), NeuroD (89 + 2) and Hes6 (52 + 4); in addition, the stem cell marker Nanog was reduced to 94 + 2 (*n* = 6; *p* < 0.01 for all; Figure 2A). Initial qRT-PCR results suggested that Wnt-11 and E-cadherin mRNA levels were inversely correlated in PDAC cells. Indeed, following the Wnt-11 siRNA treatment in Panc-1 cells, the level of E-cadherin increased to 529 + 3% (*n* = 6; *p* < 0.01). In contrast, the other three EMT markers studied all showed decreases to the following percentages: Snail (73 + 4), Vim (83 + 3) and Twist (84 + 3) (*n* = 6; *p* < 0.01 for all; Figure 2B). Taken together, all results are consistent with Wnt-11 controlling EMT [22,37] and being involved in the promotion of neuronal characteristics [16,37,38].

### 3.3. Effect of Silencing Wnt-11 on Cellular Invasiveness

The invasiveness of Panc-1 cells was studied in Boyden chambers with Matrigel (Figure 3). Cells demonstrated a noticeable invasion over 16 h. Silencing Wnt-11 resulted in significant suppression of invasiveness by 23 + 2% (*n* = 3; *p* < 0.01; Figure 3A). As expected, there was no change in the cell number over the 16 hours (Figure 3B). Thus, Wnt-11 could promote invasiveness independently of proliferation.

### 3.4. Expression in Biopsies 

We also determined the Wnt-11 mRNA levels in clinical samples of PDAC and compared these with ‘matched’ control tissues (Figure 4A). In direct agreement with the in vitro data, the average level in PDAC was significantly (four-fold) higher than the matching controls (*p* = 0.02; *n* = 111). We then questioned any association between the Wnt-11 mRNA expression and TNM stage (Figure 4B). Patient samples classified as T1, expressed the least Wnt-11. This was significantly (1931 + 125%) higher for T2 (*p* = 0.04). This increase was maintained for T3 and T4 with no further change (*p* > 0.05 for T3 and T4 cf. T2 and T3 cf. T4; Figure 4B).

## 4. Discussion

The main results are as follows: 1) Wnt-11 mRNA expression occurred commonly in PDAC cell lines and was significantly higher in PDAC compared with matched control tissues; 2) Wnt-11 promoted NEM expression and EMT as well as cellular invasiveness without affecting proliferative activity; 3) an increased level of Wnt-11 expression in PDAC patients was associated with TNM stage as a relatively early event. 

### 4.1. Wnt-11 Expression in PDAC Cells and Tissues: Control of NEMs, EMT and Invasiveness

We showed that Wnt-11 mRNA is expressed in three different PDAC cell lines. In agreement with this, Wnt-11 mRNA was present in PDAC tissues and this was at a significantly higher level compared with corresponding normal tissues. The in vitro pathophysiological role of the Wnt-11 was investigated by a siRNA approach. Thus, silencing Wnt-11 suppressed the expression of a range of biomarkers associated with metastasis, especially NEMs and those involved in EMT. Such effects suggest that Wnt-11 would promote PDAC aggressiveness and this was confirmed by Matrigel invasion assays. Interestingly, the inhibitory effect on invasion was seen without any change in proliferative activity, which is consistent with the notion that primary and secondary tumourigenesis (i.e., proliferation vs. invasion) can be controlled differently, even independently [16,39]. 

Overall, members of the Wnt family of secreted proteins are known to play a significant role in determining the “stemness”, EMT, acquisition of neuronal characteristics and invasiveness of carcinomas [16,17,20]. The level of Wnt-11 mRNA expression correlates strongly with levels of neuroendocrine differentiation in cancers of the prostate and breast [16,39]. Wnt-11 also controls expression of a neurone-specific enolase, which drives neuronal differentiation and enhanced cell viability and migration in prostate cancer [16]. These findings constitute strong evidence that Wnt-11 promotes migratory behaviour of cells, including multipotent stem/progenitor cells, during development and cancer progression, including cancers of breast, prostate, colon and cervix [16,19,39,40,41,42]. This has now been extended to PDAC in vitro and in vivo. Thus, Wnt-11 seems to play a consistent role in the pathophysiology of cancer. Indeed, Wnt-11 is evolutionarily a highly conserved protein, regulated by TGFβ and calcium in a wide variety of cell types [20,42,43]. 

### 4.2. Mechanistic Aspects 

At present, the mechanism(s) underlying the role of Wnt-11 in promoting cancer (including PDAC) cell invasiveness is/are unknown. Overall, Wnt-11 is a member of the non-canonical Wnt family, which is associated with three different signalling pathways: Planar cell polarity (PCP), Ca^2+^ and other β-catenin/TCF independent events [44]. Here, insights can be obtained by focusing on Ca ^2+^ signalling. It is known that the Frizzled family of Wnt receptors, including Wnt-11, can elicit the release of intracellular Ca^2+^ [42]. In turn, the rise in Ca^2+^ can promote invasiveness by enhancing both cellular motility and proteolysis, integral to the Boyden chamber (Matrigel) assay. For example, Ca^2+^-activated K^+^ channels were shown to enhance motility of Mia-Paca cells [45]. Another pathway could involve activation of ROCK [46,47]. As regards proteolysis, matrix metalloproteinases (MMPs) are known to be upregulated in response to non-canonical Wnt stimulation during development and tissue remodelling and promote cell migration [48,49]. Increased MMP (e.g., MMP-2, MMP-7 and MMP-9) expression occurs frequently upon Wnt stimulation of tumour cell invasion [50,51,52,53]. There is also substantial evidence for multi-modal regulation of MMPs by Ca^2+^ [53]. Consistently with the proposed non-canonical Wnt-Ca^2+^ association, it has been shown that intracellular Ca^2+^ signalling can trigger EMT induction in human cancers [54]. Moreover, TGF-β1 and Ca^2+^ have synergic effects in promoting EMT and osteochondral differentiation via Wnt-11 and L-type calcium channels, respectively [22]. Further research is required to determine the precise role of Ca^2+^ signalling in the pathophysiology of PDAC. 

### 4.3. Wnt-11 Expression in PDAC and Survival 

Our tissue analyses complement our in vitro data. Thus, in our cohort of paired human tissue samples, the levels of Wnt-11 mRNA were significantly higher in PDACs compared to their respective normal adjacent tissues. Furthermore, within this tissue set, expression was already high at TNM stage II and maintained at stages III and IV. Previous work on different cancer types also suggested that Wnt family members play a significant role in TNM staging [55,56]. These results are consistent with Wnt-11 expression being a relatively early event in PDAC development/progression. Some of the critical data obtained from the Human Protein Atlas database [34] is that it includes 176 samples from different grading stages at PDAC. All data indicate that higher-grade patients showed diminished survival analysis with up-regulated Wnt-11 expression. Five-year survival for the “higher” group was 26%, compared with 31% for the group with lower Wnt-11 expression (*p* = 0.02, Figure 5). In the first instance, these data give further credence to the results from the in vitro and tissue analyses. More broadly, as noted above, since functional Wnt-11 expression occurs in other cancers as well, its role in determining survival may be more widespread in oncology [57]. 

## 5. Conclusions

In conclusion, our data are consistent with Wnt-11 occurring early in the development of PDAC and controlling cellular expression of NEMs, EMT and invasiveness. More studies are needed to compare Wnt-11 expression and its role in non-malignant pancreatic cell lines and tissues and its function. Therefore, unsurprisingly, upregulation of Wnt-11 expression ultimately worsens patient survival. Thus, Wnt-11 may be considered as a novel prognostic biomarker for PDAC. Since this is a potential functional biomarker, inhibiting Wnt-11 expression and/or activity may also be of therapeutic importance [58].

## Figures and Tables

**Figure 1 genes-10-00921-f001:**
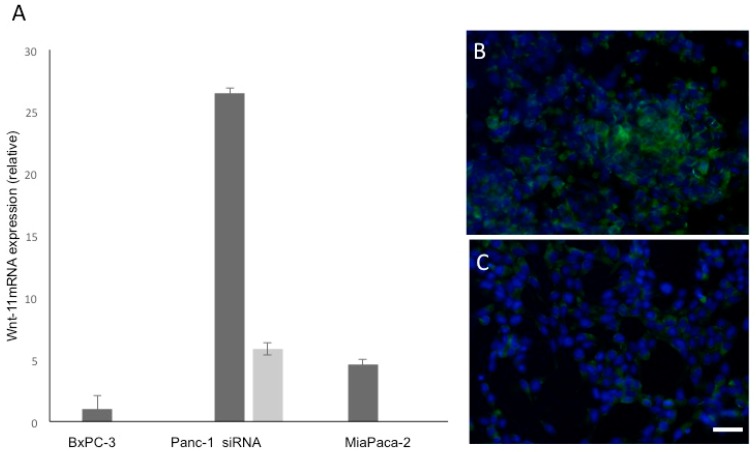
Wnt-11 mRNA relative expression profile in Bx-PC3, Panc-1 and MiaPaca-2 cells, and Wnt-11 siRNA on Panc-1 cells. (**A**) Real-time PCR result of Wnt-11 relative mRNA expression. Data are plotted as fold-differences relative to the Bx-PC3 cell Wnt-11 mRNA expression level after normalising with housekeeping gene (RPII) level. Panc-1 cells express the highest amount of Wnt-11 mRNA (folds more than Bx–PC3; *n* = 6; *p* < 0.01) among three pancreatic cancer cell lines and Wnt-11 siRNA reduces the expression 4.52-fold (light grey bar; *n* = 6; *p* = 0.011). All data were analysed as means ± standard errors. Statistical significance was determined using Student’s *t*-test or ANOVA with Newman–Keuls post-hoc analysis were used, as appropriate. (**B**,**C**) Immunostaining for Wnt-11 (green) and To-Pro (blue) in Panc-1 cells transfected with Wnt-11siRNA. (*B*) Control. (*C*) Wnt-11 siRNA for 48 hours. Scale bar, 20 µm (applicable to both panels).

**Figure 2 genes-10-00921-f002:**
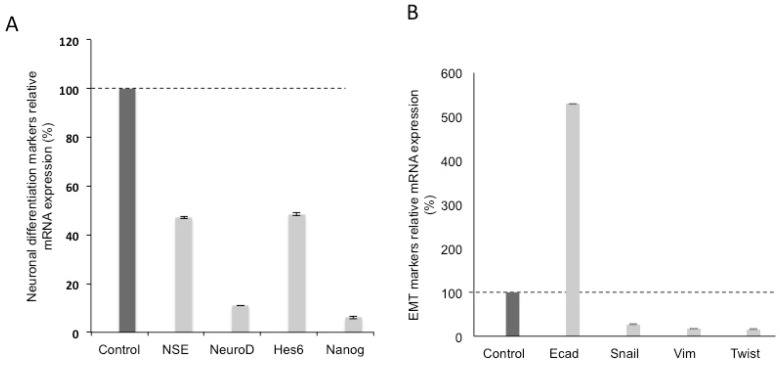
Wnt-11 promotes epithelial-mesenchymal transition (EMT) and neuronal differentiation in Panc-1 cells line. (**A**) qRT-PCR for NSE, NeuroD, Hes6 and Nanog relative mRNA expressions after transfection of Panc-1 cells either with scrambled or Wnt-11 siRNA after 48 hours. Wnt-11 siRNA. (**B**) qRT-PCR for EMT markers; Ecad, Snail, Vim and Twist relative mRNA expressions after transfection of Panc-1 cells either with scrambled or Wnt-11 siRNA after 48 hours. NSE, NeuroD, Hes6 and Nanog mRNA expressions and they were reduced after silencing Wnt-11 (53%, 89%, 51.5% and 94% respectively; *n* = 6; *p* < 0.05 for all). Wnt-11 siRNA increased Ecad mRNA expression 5.29-fold in Panc-1 cells (*n* = 6; *p* = 0.02). Other EMT markers, such as Snail, Vim and Twist expressed in Panc-1 and MiaPaca-2 cells and this expression was reduced by Wnt-11 siRNA in Panc-1 cells (Fig 2B; min 70% reduction for all, *n* = 6; *p* < 0.05). Statistical significance was determined using Student’s *t*-test and ANOVA were used.

**Figure 3 genes-10-00921-f003:**
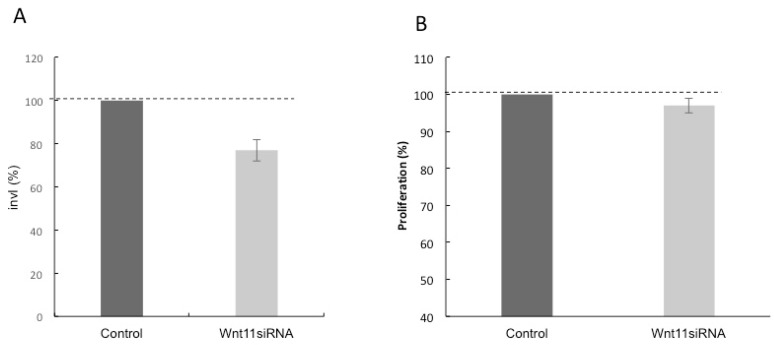
Functional evidence for Wnt-11-induced reduction of Matrigel invasion on Panc-1 cell lines. Panc-1 cells were transfected for 72 hours with control or Wnt-11 siRNAs, plated on Matrigel coated transwell filters and the extent of invasion determined after 16 hours. (**A**) Wnt-11 siRNA decreased invasion by 23% (*n* = 3; *p* = 0.02). The results are plotted as Invasion Index (InvI, %), which is the percentage of invaded cells compared to the total number of cells seeded. (**B**) The total cell number/proliferation did not change during the course of the experiment (*n* = 3; *p* > 0.05). All data were analysed as means ± standard errors.

**Figure 4 genes-10-00921-f004:**
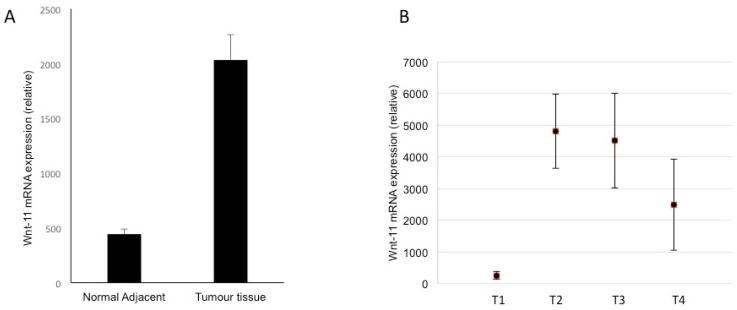
Clinical evidence that Wnt-11 was significantly upregulated in a cohort of pancreatic tumour samples. (**A**) In total, 111 pairs of tumour and adjacent control tissues were studied and Wnt-11 expression (relative) appeared to be increased in the tumour tissue compared to adjacent normal tissues (*p* = 0.02). (**B**) A high level of Wnt-11 expression (relative) was found in patients classified according to T staging (*p* < 0.05 for T1 vs T2 or T3 or T4; T2 vs T3 or T3 vs T4 *p* > 0.05). All data were analysed as means ± standard errors.

**Figure 5 genes-10-00921-f005:**
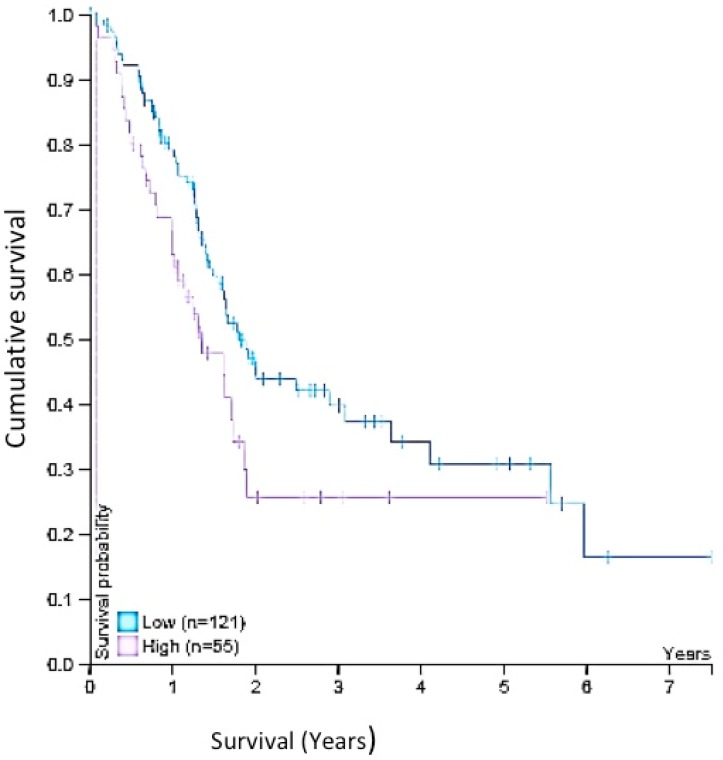
A survival curve gained from Human Protein Atlas shows that a high expression of Wnt-11 reduces cumulative survival of PDAC. (*p* = 0.02). A low expression (*n* = 121)**.** High expression (*n* = 55). 5-year survival high 26%; 5-year survival low 31%. [34]. Kaplan–Meier survival estimators examined the prognosis of each group of patients, and the survival outcomes of the two groups were compared. Bylog ranktests. https://www.proteinatlas.org/ENSG00000085741WNT11/pathology/pancreatic + cancer # Location. Accessed on 22 October 2019.

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
