# Peer review of "Wnt-11 Expression Promotes Invasiveness and Correlates with Survival in Human Pancreatic Ductal Adeno Carcinoma"

_genes, 2019, doi:10.3390/genes10110921_

Round 1

Reviewer 1 Report

The Authors describe the analysis of the expression of Wnt-11 in pancreatic ductal carcinoma cell lines and the relationship between Wnt-11 silencing and the level of expression of genes related to stemness and cell migration and the effects on cell invasion. Moreover, they provide evidence that Wnt-11 expression (mRNA level) is elevated in pancreatic cancer patients and associated with survival.

The paper is simple but well prepared and the Authors show interesting results.

However, the methodological section could be improved. When it comes to the reference gene in gene expression analysis the Authors mention cytokeratin 19 and then RNA polymerase while only RPII is mentioned in the Results section. 

What transfection agent was used for siRNA transfection? 

In the matrigel invasion assay, what were the exact experimental conditions - was there any difference in the media that were added to the upper and lower chambers? Was any chemoattractant used?

The Authors used MTT assay for the assessment of proliferation. This is not the best assay to serve this purpose. I suggest that the Authors additionally perform other proliferation assays in siRNA transfected cells - e.g. BrdU incorporation, cell cycle analysis and clonogenic assay.

The Authors report changes in the expression of genes related to stemness upon transfection with anti-Wnt11 siRNA, but did not analyze it functionally. 

Author Response

Regards,

Pinar Uysal-Onganer

Reviewer 2 Report

Nice manuscript, based on consistent and to-the-point experiments. It adds valuable knowledge to the topic and fits to this special issue of Genes.

Nevertheless I recommend substantial improvement of the manuscript as follows:

INTRODUCTION:
Use higher ranked and less special references for epidemiological and prognostic details of PDAc like your [1-4], e.g. use Siegel et al. "Cancer Statistics, 2019"; Strobel et al. "Optimizing the outcomes of pancreatic cancer surgery"
and adapt your sentence "‘conventional’ treatments, such as surgery, radiation therapy, 50 chemotherapy, and more recently immunotherapy so far have shown limited impact on progression of PDAC" to what is really said in your reference of Neoptolemos et al. [11]: that today surgery and chemotherapy do in fact make a prognostic difference for localized and locally advanced PDAC!

RESULTS:
to be changed in  the figures and associated running text.

Fig.1: description: what do the bars and antennas stand for, means ± standard errors? How many repetitions of the experiments are included here? Which statistical tests were used here for calculating differences?
Photos of 1B and 1C are of bad quality, to be replaced (maybe only in my copy).

Fig.2: How many repetitions of the experiments are included here? 2A: what does the control stand for? why do not all neuronal markers have their own control bar like in 2B? Homogenize! 2A&2B: homogenize bar design with other figs.2A&2B: description: what do the bars and antennas stand for, means ± standard errors? Which statistical tests were used here for calculating differences?

Fig.4: What do the bars/ boxes and antennas stand for, means ± standard errors etc.? Which statistical tests were used here for calculating differences?

Fig.5: Graphic is of bad quality, to be replaced (maybe only in my copy). You cite [67] as "Human Protein Atlas", I think you meant [57] like in the running text? Which statistical tests were used here for calculating differences?

The whole part about survival stratisfied by WNT-11-expressions needs clarification, in the last results-sentence of the abstract as well as in the last passage of the results-part:
where do the n=121 + 55 come from - the database (as you say in the discussion part. Reference in fig. description is wrong as described above) or your 111 patients from Beijing? Descrribe how survival data come from proteinatlas.org into your manuscript since it i not described in your reference [57]. PDACs are summarized here (treatment, patient characteristics)? And the sentence in the abstract suggest that it have been your patients.

DISCUSSION:

Name  limitations of your study and what future studies should origin from them, like for example in your in vitro part you had not tested  non-malignant pancreatic epithelial cells as baseline-comparison.

Author Response

Regards,

Pinar Uysal-Onganer

Reviewer 3 Report

The current research paper was aimed to evaluate whether Wnt-11 could play a significant role in the pathophysiology of human pancreatic ductal adenocarcinoma (PDAC).

The findings of this study could be of interest for the researchers in the field, especially in the part of clinical data. However, some points need to be addressed:

- In the introduction section, the authors should better explain their aims and hypothesis, especially the reasons to study Wnt-11 in this cancer type.

- The motive to analyse neuronal markers like NSE, NeuroD, Hes6 and Nanog for mRNA expression is not enough clear in the presentation of data and should be better explicated.

- In the paragraph “Wnt-11 mRNA expression profiling” of results’ section, the authors are too elusive when they mention RNA sequencing experiments, which are also detailed in the methods. Indeed, these data are not shown in any figure or table of the manuscript and it is not clear which RNA samples were analysed and the main obtained results. Are the raw data available online at any website? If so, the authors should cite the database and the relative accession number.

- The method to gain the survival curve from Human Protein Atlas should be mentioned (address of website, date of access, etc.).

Other points:

- The authors assess that all data were analysed as means ± standard errors and also the bar graphs are illustrated accordingly. This is in contrast with the description of their data throughout the whole text and needs to be corrected.

- In the legend of Figure 5 reference number [67] does not exist. Conceivably, it is number [57].

- In the abstract, "Pancreatic ductal carcinoma (PDAC)" should be replaced by "Pancreatic ductal adenocarcinoma (PDAC)".

Author Response

Regards,

Pinar Uysal-Onganer

Round 2

Reviewer 2 Report

Points raised by reviewers incompletely transferred.

Point 2: Fig.1:
My points have been transferred to the fig. description at the end of the manuscript not to the one in the text.

Point 3: Fig.2:
My points have not been transferred sufficiently, neither in the fig. description nor in the text. Explain how you set WNT-1 expression of Panc 1 as 100% and compared the reduction of the  other markers to that. Both in the text and fig. description. 2A&2B: homogenize bar design with other figs. Which statistical tests were used here for calculating differences?

Point 6: The part about survival stratisfied by WNT-11-expressions:
Still the patient collective used here needs to be sufficiently described inthe methods section

Author Response

Point 2: Fig.1: 
My points have been transferred to the fig. description at the end of the manuscript not to the one in the text.

Response 2: We apologise for misunderstanding the comment earlier. We now added the following to the results section. Statistical significance was determined by using Student’s t-test.

Point 3: Fig.2: 
My points have not been transferred sufficiently, neither in the fig. description nor in the text. Explain how you set WNT-1 expression of Panc 1 as 100% and compared the reduction of the  other markers to that. Both in the text and fig. description. 2A&2B: homogenize bar design with other figs. Which statistical tests were used here for calculating differences?

Response 3: We had named the statistical tests were used in the methods section.We now added the following to the results section. Control transfection that was used as a ‘baseline’ and presented as 100 % and statistical significance was determined by using Student’s t-test. We homologised the figs.

Point 6: The part about survival stratisfied by WNT-11-expressions:
Still the patient collective used here needs to be sufficiently described in the methods section.

Response 6: We were unsure about this comment. The survival data came from the Protein Atlas website, we included info on the methods section.  We have now included an extra section called ‘survival analysis’ on the methods section.

Survival data was obtained on 22nd of Oct 2019 from Human Protein Atlas website:

https://www.proteinatlas.org/ENSG00000085741-WNT11/pathology/pancreatic+cancer#Location

Wnt-11 expression of total 176 patients were analysed, (low expression; n=121 and high expression; n=55) and Kaplan-Meier survival estimators examined the prognosis of each group of patients, and the survival outcomes of the two groups were compared by log-rank tests [34].

We have also added extra info on discussion: One of the critical data obtained from the Human Protein Atlas database [34], is that it includes 176 samples from different grading stages at PDAC. All data provides that higher grade patients showed diminished survival with upregulated Wnt-11 expression.